# The Effects of COVID-19 Lockdown on the Perception of Physical Activity and on the Perception of Musculoskeletal Symptoms in Computer Workers: Comparative Longitudinal Study Design

**DOI:** 10.3390/ijerph19127311

**Published:** 2022-06-14

**Authors:** Sara Moreira, Maria Begoña Criado, Maria Salomé Ferreira, Jorge Machado, Carla Gonçalves, Cristina Mesquita, Sofia Lopes, Paula Clara Santos

**Affiliations:** 1ICBAS, Instituto de Ciências Biomédicas Abel Salazar, Universidade do Porto, 4099-002 Porto, Portugal; jmachado@icbas.up.pt; 2ESS IPVC, Escola Superior de Saúde, Instituto Politécnico de Viana do Castelo, 4900-314 Viana do Castelo, Portugal; salomeferreira@ess.ipvc.pt; 3CBSin—Center of BioSciences in Integrative Health, 4000-105 Porto, Portugal; mbegona.criado@ipsn.cespu.pt; 4TOXRUN—Toxicology Research Unit, University Institute of Health Sciences, CESPU, 4585-116 Gandra, Portugal; 5UICISA: E—Health Sciences Research Unit: Nursing, Nursing School of Coimbra (ESEnfC), Portugal School of Health, Polytechnic Institute of Viana do Castelo, 4900-314 Viana do Castelo, Portugal; 6LABIOMEP—Laboratório de Biomecânica do Porto, Universidade do Porto, 4200-450 Porto, Portugal; 7ESDL IPVC, Escola Superior Desporto e Lazer, Instituto Politécnico de Viana do Castelo, Rua Escola Industrial e Comercial de Nun’Álvares, 4900-347 Viana do Castelo, Portugal; carlagoncalves@esdl.ipvc.pt; 8Research Center in Sports Performance, Recreation, Innovation and Technology (SPRINT), 4960-320 Melgaço, Portugal; 9ESS PPorto—Department of Physiotherapy, School of Health, Polytechnic of Porto, António Bernardino de Almeida, 400, 4200-072 Porto, Portugal; ctmesquita@ess.ipp.pt (C.M.); srl@ess.ipp.pt (S.L.); paulaclara@ess.ipp.pt (P.C.S.); 10CIR—Centro de Investigação e Reabilitação, ESS P, 4200-072 Porto, Portugal; 11CESPU—Departamento de Tecnologias de Diagnóstico e Terapêutica, Escola Superior de Saúde do Vale do Sousa, Instituto Politécnico de Saúde do Norte (IPSN), 4585-116 Paredes, Portugal; 12CIAFEL—Research Center in Physical Activity, Health and Leisure, Faculty of Sport, Laboratory for Integrative and Translational Research in Population Health (ITR), University of Porto (FADEUP), Rua Dr. Plácido Costa, 91, 4200-450 Porto, Portugal

**Keywords:** occupational health, musculoskeletal symptoms, physical activity level, homeworking, computer workers

## Abstract

Lockdown resulting from the pandemic led to a change in the health habits of the computer workers community. Sedentary work, together with less active lifestyles, aggravated by the COVID-19 pandemic leads to impacts on physical activity (PA) and can contribute to the development of musculoskeletal symptoms (MSS). **Aim(s):** Understand the effects of lockdown on the perception of physical activity levels and on the perception of frequency of musculoskeletal symptoms, over periods of 12 months and 7 days, in computer workers. **Methods:** Longitudinal comparative study between 2019 (M1) and 2021 (M2), over 18 months, in 40 volunteer participants. The inclusion criteria were full-time workers aged between 18 and 65 and the exclusion criteria included diagnosis of non-work-related medical conditions. In addition to a socio-demographic questionnaire, the Nordic musculoskeletal questionnaire (NMQ) was used to evaluate the MSS and the International Physical Activity Questionnaire (IPAQ), was used to analyse the perception of the level of PA. These questionnaires were used in two assessment stages (M1 and M2). McNemar test and Wilcoxon paired test were used to evaluate the effect of lockdown on the perception of PA, and on the perception of frequency of musculoskeletal symptoms. **Results:** The MSS prevalence in the previous 12 months increased significantly in the neck (M1: 45.0%, M2: 62.5%, *p* = 0.046), in the shoulders (M1: 37.5%, M2: 55.0%, *p* = 0.033), and in the hands/wrists (M1: 25.0%, M2: 45.0%, *p* = 0.019). The mean pain score increased in the shoulders (1.43 ± 2.24, 2.35 ± 2.55, *p* = 0.003) and in the elbows (0.18 ± 0.59, 0.60 ± 1.34, *p* = 0.015). No differences were found in the PA between M1 and M2, but the weekly mean sitting time increased from 4.75 ± 2.26 to 6.26 ± 2.65 (*p* < 0.001). **Conclusion:** After 18 months it became clear that MSS perception increased mainly in the neck, shoulders and hands/wrists with a significant increase in pain intensity in the shoulder and elbow regions. The weekly sitting time increased significantly. Further studies are needed in order to determine the impact of teleworking in a pandemic context. But multifactor behind these results should be taken into account by health institutions and those responsible for the Prevention of Occupational Risks in Computer Workers in order to adopt educational strategies for the promotion of Physical activity (PA), in these workers.

## 1. Introduction

New technologies contributed not only to an improvement in working conditions, but also to the development of occupational and health problems. In this sense, the number of workers using computers for most of their working hours continues to grow at an ever-increasing rate [1] and results of various studies pointed out the presence of specific health symptoms associated with the use of computers [2,3].

The World Health Organization has characterized work-related illnesses as multifactorial, including physical, organisational, psychosocial, individual and sociocultural factors. Work-related musculoskeletal disorders have grown substantially in recent decades and are the most important causes of absence and disability at work [4,5,6]. 

Office workers, or computer workers (CWs), are considered to be those who work for approximately two-thirds of their working hours on primary tasks that generally involve the use of computers and a sitting position [7,8,9]. These workers are therefore at increased risk of a number of chronic diseases due to their sedentary behaviour [10,11].

Several risk factors may be associated with the development of musculoskeletal symptoms (MSS). The ones reported in the literature are repetitive activities, maintained postures, long working hours, inadequate furniture and lack of conditions to perform the work. Repetition of movements, maintained positioning and lack of pauses are common in computer workers [4,5]. This sedentary work leads to poor postures, flexion postures adopted during long periods of work, associated with repetition of tasks, causing greater tension in muscle and ligament structure, thus increasing the prevalence of work-related musculoskeletal injuries [5,8,10,12,13,14,15,16]. According to several studies, the cervical and lumbar regions are the most affected, followed by disorders in at least one region of the upper limbs [2,3,8,17,18,19,20,21,22]. 

Remaining in the sitting posture, associated with the lack of postural variability, generates several changes in the musculoskeletal structures: it increases the internal pressure in the nucleus of the intervertebral disc, stretches all the structures of the spine, reduces the return circulation of the lower limbs and promotes the development of inflammatory processes in bone and muscle structures with associated painful symptoms [23,24,25]. Other studies concluded that long working hours with little or no interruption are risk factors for triggering pain in the cervical region, shoulders and other segments of the spine [4,26].

Some studies identified the great impact of work-related MSSs in economic terms (loss of productivity and higher social expenditure). In Germany for example, musculoskeletal and connective tissue disorders accounted for EUR 17.2 billion of production loss in 2016 [27].

Among the therapeutic approaches to decrease MSS, several authors recommend the regular practice of Physical Activity (PA), which is defined by the WHO as any body movement produced by the muscles that requires energy expenditure. It is observed that a large part of the population that has MSS does not perform PA regularly, with the justification of lack of time, work overload and lack of motivation, among others [4,6,28].

December 2019 saw the appearance of COVID-19, a communicable disease caused by infection of SARS-CoV-2 coronavirus [29]. This health condition spread, affecting several countries, leading the WHO to declare a pandemic that continues today. As such, mandatory teleworking was ordered to prevent transmission. The COVID-19 pandemic has shaken the structures of contemporary society and required a reorganisation of companies and societies in general. Teleworking, home office, was already a practice in some companies and, with the new coronavirus, it became mandatory for all [30].

The need for a lockdown led to a fundamental change in the lifestyles of the entire population. Although this strategy proved to be effective in combating the pandemic, the quarantine had significant effects on other relevant aspects of the health of the population that was subject to this lockdown [31,32]. Restrictions on movement and the ban on carrying out activities in outdoor spaces inevitably disrupted everybody’s routines, increasing the likelihood of individuals experiencing reduced levels of physical activity, an increase in immobility, anxiety and depression, which significantly affects health and increases several health risk factors [33].

This meant that workers’ homes had to become a place of work, education and leisure. Teleworking has some benefits, such as a better family-work balance, reduced fatigue and increased productivity [34]. However, factors such as a lack of clear definition of physical and organisational boundaries between work and home, extended working hours and limited support from organisations can negatively impact the physical and mental health of workers [34], leading to a decrease in health-related quality of life [35,36]. Against this background, recent reviews investigating the effects of physical exercise on the health of office workers [22,37,38] reported significant and protective effects of physical exercise on musculoskeletal pain symptoms (i.e., neck pain and low back pain) [22,38,39]. Recent studies carried out during the pandemic also suggested that regular physical activity can be an accessible auxiliary tool for the immune system against possible COVID-19 infection [40].

Improving PA to reduce MSS is therefore a challenge for occupational health, because the work context is crucial for the development and creation of health promotion actions within the scope of work activity [38,41,42]. In this context, it is extremely important to investigate the perception of PA and perception of MSS that computer workers have and how to change attitudes in order to introduce successful programmes to promote health-oriented physical activity in a variety of social groups [42,43,44]. 

The aim of this study was to compare (i) the frequency of musculoskeletal symptoms and (ii) the perception of physical activity in computer workers at an automotive sector company in northern Portugal, between the pre-pandemic and pandemic periods. In addition, specific objectives were defined to verify the difference in the frequency of symptoms at the two evaluation moments; to compare the most affected anatomical regions between the two periods; to investigate the difference in the pain score in those anatomic regions in the pre-pandemic and pandemic periods related to the weekly sitting time in both moments. As such, the study hypothesis was that lockdown led to an increase in the perception of musculoskeletal symptoms and the perception of physical activity in computer workers.

## 2. Materials and Methods 

### 2.1. Study Design

A comparative longitudinal study was performed with two data collection moments, the first one from 5th to 12th of June 2019, and the second between 10th of December 2020 and 29th of January 2021. In addition, our study can be considered single blind, since the data were processed by an independent observer, after self-completion online, and analysed by a different group of observers. 

### 2.2. Ethical Procedures

Data were collected from adult CWs. Ethical approval for this study was obtained from the Ethics Committee at the Abel Salazar Institute of Biomedical Sciences CHUP/ICBAS (963). All the participants were informed about the study aims and procedures and they provided written consent for their participation. The participants could refuse to participate in the study at any point under Law 67/98 of 26 October 1998 (Law on the Protection of Personal Data (transposing into the Portuguese legal system Directive 95/46/EC of the European Parliament and of the Council of 24 October 1995 on the protection of individuals with regard to the processing of personal data and on the free movement of such data) and the World Medical Association Declaration of Helsinki Ethical Principles for Medical Research.

### 2.3. Sample Recruitment and Elegibility Criteria

The target population of this study consisted of 424 adult CWs at an automotive sector company in northern Portugal. 

Defined inclusion criteria were: full-time employees aged between 18 and 65 who had signed the informed consent. The exclusion criteria included diagnosis of non-work-related medical conditions, such as: ankylosing spondylitis, chronic joint diseases, neurological diseases, relevant (osteoarticular) surgeries, significant artificial joint replacement, articulation, multiple sclerosis, myotonic dystrophy or neurodegenerative diseases and congenital malformations of the musculoskeletal system [20,45].

Before the study questionnaire was applied, a visit to the company was made in order to establish initial contact with the possible participants and their work environment.

The study questionnaires were entered into the *Google Forms* platform to make them faster and more accessible for the 424 computer workers to complete. The questionnaire *link* was provided to the potential participants via *email* by the company’s human resources management to safeguard the confidentiality of the participants’ personal *emails*. Previously, all the computer workers were informed by the company’s human resources department about the objectives of the study and, after that, the questionnaires were presented. Subsequently, time was allotted to clarify any doubts and questions that remained. The forms were filled out in two stages. At a first moment (M1) the questionnaire was sent in June of 2019 to the company’s 424 workers. The second moment (M2) was between December 2020 and January 2021. A total of 119 responses were obtained, 304 individuals refused to participate and 1 participant was excluded for having a diagnosis of Ankylosing Spondylitis, thus forming a final non-probabilistic sample of 40 computer workers (Figure 1).

### 2.4. Questionnaire for Collecting Data

The final questionnaire consisted of three parts. One with sociodemographic questions, a second with the Nordic Musculoskeletal Questionnaire (NMQ-*vPt*) [46] and a third with the International Physical Activity Questionnaire—short version [47]. A pilot study was carried out to test the procedures. The survey was designed to take no longer than 15 min to complete and it was a self-administered questionnaire.

### 2.5. Instruments

#### 2.5.1. Sociodemographic Questionnaire

This questionnaire was designed by the main researcher to characterise the sample and collect sociodemographic data [44]. General information such as sex, birth date, relationship status and education, anthropometric variables, such as weight and height, were self-reported, followed by questions about participants’ medical history and lifestyle and, lastly, work-related questions. The Cronbach’s Alpha of the sociodemographic questionnaire was 0.72—it was calculated through the “optimal scaling” procedure implemented in SPSS. To capture additional contextual information, the number of hours per week in domestic and leisure activities and sitting time working were included. 

#### 2.5.2. Nordic Musculoskeletal Questionnaire (NMQ-*vPt*)

The NMQ is validated for the Portuguese population and facilitates the evaluation of musculoskeletal symptoms [46]. It consists of questions related to nine anatomical regions (neck, shoulders, elbows, wrist, thoracic region, lumbar region, hips, knees, ankles and feet). The individual is asked if he/she had any symptoms in the last 12 months and in the last 7 days in one or more regions and if in the last year he/she has had to avoid his/her normal day to day activities due to symptoms. The anatomical regions are highlighted on a *body chart* so that there is no doubt about the area they are referring to [48]. The questionnaire also presents a numerical pain scale (NPS) that allows the study participant to classify their pain in the “last 7 days” according to the indicated regions.

This questionnaire has moderate criterion validity and good reliability and is validated for the Portuguese population [46]. The questionnaire presents psychometric values for reliability; test-retest using the Kappa correlation coefficient ranged between 0.677 and 1. The internal validity of the questionnaire was verified by the Kuder-Richarson correlation coefficient, which was good (0.855) [46].

#### 2.5.3. International Physical Activity Questionnaire (IPAQ)–Short Version 

The short version IPAQ makes it possible to evaluate the perception of PA level and sedentary behaviour. It is part of an international effort to find an instrument that may be used worldwide in order to determine the level of physical activity of several populations. This instrument has been validated for several countries simultaneously. The validity and reliability of questionnaire published by Craig et al. (2003) has the main objective of developing a self-reported measure of physical activity suitable for assessing population levels of physical activity across countries including for the Portuguese population [47,49]. It consists of 9 questions and provides information on sedentary activity time, vigorous and moderate PA and walking time. Any PA that the participants perform, either at work or on household chores and during their free time, can be included [50].

For the IPAQ data analysis, the Guidelines for data and processing of international physical activity questionnaire were used [51]. These guidelines provide reference metabolic equivalent values (MET). Walking corresponds to 3.3 METs, moderate PA to 4.0 METs and vigorous PA to 8.0 METs. They also have formulas to quantify the total PA per minute/week. The final score (MET minute/week) is the sum of the scores of each PA level:
Walking METs = 3.3 × minutes’ walk × number of days walkingModerate METs = 4.0 × minutes of moderate activity × number of days doing moderate activityVigorous METs = 8.0 × minutes of vigorous activity × number of days doing vigorous activityTotal METs = METs walkingMETs of moderate activity + METs of vigorous activity.

Each participant is classified into 1 of 3 categories according to the final score; low, moderate and high level of PA. If the participant does not meet the conditions to enter the other categories, he/she is classified as low PA.

To fall within the moderate PA level category, a participant must meet one of the following criteria: having 3 or more days of vigorous activity for at least 20 min a day or 5 days or more of moderate activity and/or walking for at least 30 min a day, or 5 or more days of any combination of walking, moderate or vigorous intensity activities that reach a minimum of at least 600 MET-minutes/week.

Finally, to qualify for a high level of PA the individual needs to meet one of the following criteria: have vigorous intensity activity on at least 3 days reaching a minimum of 1500 MET-minute/week of total PA; or 7 days of any combination of walking, moderate intensity or vigorous intensity activities achieving a minimum PA of 3000 MET-minute/week.

Issues related to sedentary behaviour are not included in the calculation of minute/week MET, as these were developed as separate indicators [51].

This questionnaire, according to [50] has a criterion validity of r = 0.30 and a reliability of r = 0.76 (95% CI 0.73–0.77). To calculate the score, an open access *Excel* document developed by Dr Hoi Lun Cheng [52] was used. No cases were excluded during the processing of IPAQ data, following the IPAQ guidelines [51]. 

### 2.6. Statistics

Mean and standard deviation were used for describing the continuous variables, and absolute and relative frequencies were used for categorical variables. 

For the comparison between M1 and M2 (paired samples), the McNemar test and the paired Wilcoxon test were used for binary variables and for continuous variables, respectively. The effect size of the differences of continuous variables was evaluated with Cohen’s d [53] (d = 0.20 small effect, d = 0.50 medium effect, d = 0.80 large effect). A significance level of 5% was considered for the inferential tests. 

Data analysis was performed with IBM SPSS Statistics^®^, version 27.0 (Statistical Package for the Social Sciences ^®^, IBM Corp Armonk, NY, USA) [54].

## 3. Results

### 3.1. Sample Profile

Table 1 shows the sample characteristics in M1. Sample included 40 computer workers aged from 25 to 50 (M = 35.75, SD = 7.32), mostly men (65.0%), married/cohabitation (55.0%), and with secondary education (35.0%) or a Bachelor′s degree (45.0%). Half of the sample was overweight: 30.0% were pre-obese and 20.0% were obese. 

### 3.2. Musculoskeletal Symptomatology

Table 2 shows the results of musculoskeletal symptomatology in M1 and in M2. 

Results comparing M1 and M2 showed that the percentage of computer workers with symptoms in the past 12 months increased in the neck (45.0% in M1, 62.5% in M2, *p* = 0.046), shoulders (37.5% in M1, 55.0% in M2, *p* = 0.033), elbows (7.5% in M1, 20.0% in M2, *p* = 0.063), and wrists and hands (25.0% in M1, 45.0% in M2, *p* = 0.019). 

Regarding the activities of daily life limitations in the past 12 months, the only increase found was in the wrists and hands (2.5% in M1, 12.5% in M2, *p* = 0.063), with significant differences at a 10% level.

As for the symptoms in the past 7 days, the percentages increased in the neck (22.5% in M1, 37.5% in M2, *p* = 0.073) and in the shoulders (10.0% in M1, 25.0% in M2, *p* = 0.055), with significant differences at a 10% level. 

The mean pain intensity increased significantly in two regions: the shoulders (M1: M = 1.43, SD = 2.24; M2: M = 2.35, SD = 2.55; p = 0.003; d = 0.38) and the elbows (M1: M = 0.18, SD = 0.59; M2: M = 0.60, SD = 1.34; *p* = 0.015; d = 0.015).

### 3.3. Physical Activity and Sitting Time

Regarding physical activity and sitting time, Table 3 shows that there were no differences in the physical activity indicators (METs, physical activity level, meeting the WHO recommendations) between M1 and M2. 

The mean sitting time on weekdays increased significantly from 4.85 (SD = 3.05) in M1 to 6.75 (SD = 3.41) in M2 (*p* < 0.001, d = 0.59). There was also an increase in the mean sitting time on weekend days (M1: M = 4.48, SD = 2.59; M2: M = 5.03, SD = 2.61), however, the differences were not significant (*p* = 0.064), and the effect size of the differences was small (d = 0.21). 

## 4. Discussion

This study evaluated the effects of lockdown on the perception of physical activity, sitting time levels and perception of frequency of musculoskeletal symptoms over periods of 12 months and 7 days in CWs. 

Comparing the pre-pandemic and pandemic results regarding the practice of PA, it was found that there was a decrease in the percentage of workers classified as having a medium category of PA and an increase in the percentage of workers classified as having a vigorous category. However, this change was not enough to meet the PA standards recommended by the WHO [6]. These changes in PA levels may be explained by the isolation measures imposed by the pandemic, as people could only leave the house to perform PA, with a number of individuals opting for activities with higher metabolic expenditure (such as running).

A growing number of studies from the COVID-19 period reported that the limitations imposed by the pandemic lockdown are associated with significant negative changes in physical activity habits [55,56]. Also, eating behaviours and lifestyle were affected by the lockdown, with increased consumption of unhealthy food, which may have a possible negative influence on immune response [31]. Negative psychological outcomes like increasing levels of depression, anxiety and stress were also observed during lockdown [57]. Understanding the possible association between these factors and decreasing physical activity habits will be important for the development of further interventions. 

According to the literature, CWs are prone to MSS in various body regions [58,59,60,61,62,63] and regular PA practice has been shown to be a beneficial strategy to reduce the onset of MSS and lead to a reduction in pain intensity [64,65,66,67,68,69]. In previous works by our group, we also found that MSS were less frequent in CWs who followed PA recommendations [69] and we also observed a large effect of PA in the reduction of pain in CWs [68]. Additionally, PA practice can result in lower BMI, body fat percentage and blood pressure, as well as better job performance [64]. The increased systemic circulation and vasodilator capability caused by PA [10,28,58,65,70] could be one explanation for the reduction in MSS rates in this group. It is also important to remember that PA should include activities for all body parts to prevent worker’s MSS [69].

The average sitting time on weekdays increased significantly in the pandemic period. Teleworking has been associated with more time spent in sedentary activities [71,72]. In this sense our findings agree with recent studies in which the need to work from home has been shown to lead to more hours spent sitting, in online meetings and working [72,73]. The sudden change to having to work from home caused a decrease in ergonomics and relative comfort in the workspace. As in other studies, our analysis suggests that this fact may be associated with an increase in MSS during the period of lockdown [66]. 

The bodily discomforts generated by work can affect any individual who follows a work pattern where activities and tasks are poorly performed, due to the poor adaptation of the environment and the work routine to the individual capabilities and characteristics of each worker. Several studies indicate that the presence of symptoms in different body regions is due to the flexion postures adopted during long periods of work, associated with the repetition of the task, causing greater tension in the muscular and ligament structures [40,41,42]. Inadequate postural habits, such as those imposed by the sitting position and the repetitiveness of certain movements, such as CW’s, act on the human organism as an overload and are capable of leading its various defence mechanisms to compensatory actions [74]. In the sitting position, the weight of the body exerts significant pressure on the vertebral column, causing the water contained in the gelatinous substance of the nucleus to exit through the orifices of the vertebral plateau towards the centre of the vertebral bodies. Maintaining this type of posture for a long time makes the core less hydrated and thick at the end of the day. During the night, with rest, the pressure exerted on the disc decreases considerably, due to the body being relaxed. At this moment, the opposite occurs, that is, the core attracts water, returning to its initial thickness at the end of the night. However, for the disc to return to its normal thickness, a significant period of rest is necessary [74].

However, the form of work is not necessarily the only determinant of the causality of these symptoms, it is necessary to take into account other multifactor, including physical and psychological factors present inside and outside the work environment [75,76].

Regarding the most affected anatomical regions, in both moments of analysis, the regions that reported increased symptoms in the last 12 months were neck, shoulders, elbows, wrists and hands; these findings were also reported in previous studies [65]. As previously mentioned, staying in the sitting position for long periods is a risk factor for the increase of this symptomatology [65]. 

We also found that those who had symptoms in the last 7 days in the aforementioned regions (as well as in the other variables of the Nordic questionnaire) had a higher average number of sitting hours per week in relation to individuals who did not report the presence of these symptoms. Another factor that may have led to the worsening of symptoms and that was not taken into account in this study are the ergonomic issues of the work environment at home, because, although the technical gesture has not changed, the organisation of the work space at home is different and can be less adaptable to the needs of workers compared to the conditions offered at this company [71].

### Study Limitations

One of the limitations has to do with the low number of participants in the study meaning that the sample is not representative of the population and therefore has no external validity. Also, the fact that the gender was mostly male, which led to heterogeneity between the groups. Furthermore, it was not possible to verify whether the workers answered the questionnaire themselves as it was answered online. As such, many personal variables, including physiological, behavioural and psychological factors, may influence motivation to join physical activity programmes. 

As such, to make physical activity part of the daily life of CWs, it will be important to understand common barriers to physical activity and to create strategies to overcome them. Regarding the applied assessment, both biases are dual issues, taking into account that the questions referred to the past. As for the population under study, it belongs to a specific population of a company that has adopted strategies intended to promote the health of its workers by applying various health promotion actions.

In future works it will be important to follow up these workers in order to analyse whether the new physical habits resulting from lockdown and teleworking are maintained. We also suggest the analysis of variables such as sleep and the ergonomics of the workspace that were not taken into account in this study and are referred to as risk factors for MSS symptoms. In addition, more objective PA assessment measures (such as a pedometer) could be introduced to obtain more reliable data from the practice of PA.

Considering the well-established link between physical activity and health, it is likely that persistent or increased physical inactivity has medium or long-term implications for people’s physical and mental health, as well as for the quality of life of each individual. More occupational health efforts should be made to boost the level of physical activity in the workplace, in particular for groups at higher risk of inactivity or reduced physical activity, such as computer workers. Physically active people are more content and more alert [31]. There is evidence in the literature that exercise is beneficial for mental health; it reduces depression, negative moods and anxiety, and improves self-esteem and cognitive functioning [77].

Our study contributes to increasing the knowledge of this important topic, giving support for occupational risk prevention of MSSs. In line with the results obtained in this study, we recommend the introduction of active health education strategies to train workers who are involved in jobs with repetitive tasks, to increase PA levels, and greater efforts to prepare the home environment as a more comfortable and ergonomic workplace for CWs. These factors can play an important role in a healthier transition to working from home. 

## 5. Conclusions

The main strength of this study was evaluating the impact of lockdown on the perception of physical activity, sitting time levels and the perception of frequency of musculoskeletal symptoms over periods of 12 months and 7 days in computer workers.

We conclude that MSS perception increased mainly in the neck, shoulders and hands/wrists with a significant increase in pain intensity in the shoulder and elbow regions. It was clear that weekly sitting time increased significantly. Multifactor behind these results must be taken into account by health institutions and those responsible for the Prevention of Occupational Risks in Computer Workers. These findings are also important as they can be useful to adopt educational strategies for the promotion of Physical activity in the workplace, reflecting a positive perception of health in these workers. 

More studies are needed to understand the effects of the lockdown on CWs, but it seems clear that there is a need to introduce exercise programmes to promote physical activity, as this is a health promoting element for the prevention of MSS in computer workers. In this context, further analysis is needed to confirm the observed results, but there seems to be a clear need to direct efforts and resources towards improving computer workers’ PA by boosting CWs’ motivation to join exercise programmes in the workplace and introducing suitable exercise programmes and policies, with the consequent social, economic and environmental impacts on physically active, healthy populations.

## Figures and Tables

**Figure 1 ijerph-19-07311-f001:**
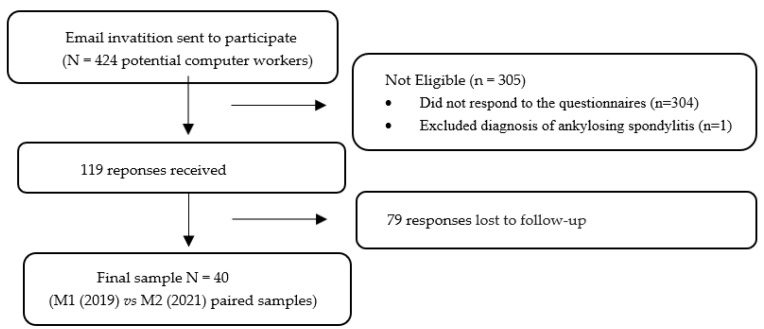
Flow diagram of the recruitment process.

**Table 1 ijerph-19-07311-t001:** Sample characteristics in M1 (*n* = 40).

	*n*	%
**Gender**		
Female	14	35.0
Male	26	65.0
**Marital Status**		
Single	15	37.5
Married/Cohabitation	22	55.0
Separated/Divorced	3	7.5
**Educational Attainment**		
Middle school	1	2.5
Secondary school	14	35.0
Bachelor’s degree	18	45.0
Master’s degree	7	17.5
**BMI Category**		
Low weight	1	2.5
Normal weight	19	47.5
Pre-obesity	12	30.0
Obesity	8	20.0
	**Mean**	**Standard Deviation**
Age (years)	35.75	7.32
Height (m)	1.72	0.10
Body mass (kg)	76.58	16.86
BMI (kg/m^2^)	25.72	4.97

BMI—body mass index.

**Table 2 ijerph-19-07311-t002:** Musculoskeletal symptomatology, limitation in daily activities and pain intensity in M1 and M2 (*n* = 40).

	Musculoskeletal Symptoms												
Anatomical Regions	Symptomatology Last 12 Months			ADL Limitation Last 12 Months			Symptomatology Last 7 Days			Pain Intensity			
	M1	M2		M1	M2		M1	M2		M1	M2		
	*n* (%)	*n* (%)	*p* ^(1)^	*n* (%)	*n* (%)	*p* ^(1)^	*n* (%)	*n* (%)	*p* ^(1)^	M(SD)	M(SD)	*p* ^(2)^	d
Neck	18 (45.0)	25 (62.5)	0.046	7 (17.5)	6 (15.0)	0.500	9 (22.5)	15 (37.5)	0.073	1.83 (2.26)	2.23 (2.42)	0.156	0.17
Shoulders	15 (37.5)	22 (55.0)	0.033	2 (5.0)	6 (15.0)	0.109	4 (10.0)	10 (25.0)	0.055	1.43 (2.24)	2.35 (2.55)	0.003	0.38
Elbows	3 (7.5)	8 (20.0)	0.063	0 (0.0)	1 (2.5)	0.500	0 (0.0)	3 (7.5)	0.125	0.18 (0.59)	0.60 (1.34)	0.015	0.41
Wrists and hands	10 (25.0)	18 (45.0)	0.019	1 (2.5)	5 (12.5)	0.063	2 (5.0)	2 (5.0)	0.750	0.65 (1.19)	0.93 (1.56)	0.183	0.20
Thoracic	5 (12.5)	4 (10.0)	0.500	1 (2.5)	3 (7.5)	0.250	1 (2.5)	0 (0.0)	0.500	0.38 (1.21)	0.20 (0.56)	0.273	0.19
Low back	22 (55.0)	24 (60.0)	0.395	6 (15.0)	7 (17.5)	0.500	10 (25.0)	13 (32.5)	0.291	2.10 (2.72)	2.38 (2.48)	0.330	0.11
Hips and thighs	11 (27.5)	8 (20.0)	0.291	1 (2.5)	1 (2.5)	0.750	2 (5.0)	2 (5.0)	0.750	0.68 (1.49)	0.48 (1.04)	0.363	0.16
Knees	14 (35.0)	16 (40.0)	0.387	3 (7.5)	2 (5.0)	0.500	5 (12.5)	5 (12.5)	0.656	1.13 (2.03)	1.00 (1.52)	0.367	0.07
Ankles and feet	12 (30.0)	9 (22.5)	0.304	1 (2.5)	2 (5.0)	0.500	2 (5.0)	4 (10.0)	0.313	0.70 (1.56)	0.50 (1.40)	0.280	0.13

*n*—absolute frequency; %—relative frequency; M—mean; SD—standard deviation; ADL—activities of daily life; d—Cohen´s d; ^(1)^ significant value of McNemar test; ^(2)^ significant value of paired Student’s *t*-test.

**Table 3 ijerph-19-07311-t003:** Physical activity and sitting time in M1 and M2 (*n* = 40).

	M1	M2	p	d
**Physical Activity (METs)**	M (SD)	M (SD)		
Vigorous (minutes/week)	791.00 (1252.18)	1044.00 (1739.67)	0.291 ^(1)^	0.17
Moderate (minutes/week)	340.00 (576.30)	279.50 (519.69)	0.241 ^(1)^	0.11
Walking (minutes/week)	526.76 (594.86)	727.24 (925.81)	0.281 ^(1)^	0.26
Total (minutes/week)	1657.80 (1825.07)	2050.78 (2675.69)	0.475 ^(1)^	0.17
**Physical Activity Level**	*n (%)*	*n (%)*		
Low	16 (40.0%)	16 (40.0%)	0.372 ^(1)^	
Medium	15 (37.5%)	13 (32.5%)		
High	9 (22.5%)	11 (27.5%)		
**Meets WHO recommendations**	*n (%)*	*n (%)*		
No	21 (52.5%)	22 (55.0%)	0.500 ^(2)^	
Yes	19 (47.5%)	18 (45.0%)		
**Sitting time**	*M (SD)*	*M (SD)*		
Weekday (hours/day)	4.85 (3.05)	6.75 (3.41)	< 0.001 ^(1)^	0.59
Weekend day (hours/day)	4.48 (2.59)	5.03 (2.61)	0.064 ^(1)^	0.21

*n*—absolute frequency; %—relative frequency; M—mean; SD—standard deviation; ADL—activities of daily life; d—Cohen’s d; ^(1)^ significant value of paired Wilcoxon test; ^(2)^ significant value of McNemar test.

## Data Availability

The data presented in this study are available on request from the corresponding author. The data are not publicly available as the confidentiality agreement reached with the company does not allow public disclosure of the data available.

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
