# Peer review of "The Effects of COVID-19 Lockdown on the Perception of Physical Activity and on the Perception of Musculoskeletal Symptoms in Computer Workers: Comparative Longitudinal Study Design"

_ijerph, 2022, doi:10.3390/ijerph19127311_

Round 1

Reviewer 1 Report

I don't understand why at the beginning of the summary they say that the study is done in 12 months and 7 days and at the end in the conclusion, they talk about 18 months, they should clarify it.

Also in the summary they include the acronym AF the first time the term physical activity appears and then they do not use the acronym when the term appears (4 times) they should fix it.

The sample is not very significant (40 subjects), although it may be possible to accept it due to the instruments they use, but they must state it. It would also be convenient for them to say in which country the study was carried out so that it can be replicated in other places.

All these aspects included in the abstract are repeated in the article, so it is suggested that it be reviewed before accepting it.

Reviewer 2 Report

The paper entitled “The effects of COVID-19 lockdown on the of perception physical activity and on the perception of musculoskeletal symptoms in computer workers: Comparative Longitudinal Study Design” is consistent with the profile of the Journal.

1.       The paper covers up-to-date scientific topic related to COVID pandemic with high originality and significance of content.

2.       Abstract is well prepared. Just one remark IPAQ shall be written International Physical Activity Questionnaire.

3.       The introduction explains thoroughly the scientific background. Therefore I am convinced why this context is important. The authors use up-to-date literature to present the discussed problem in the paper.

4.       Material and methods section is well prepared. The information provided in sections 2.1.-2.5 is clearly presented.

Regarding point 2.4.3. The authors provide information on the validity and reliability of the English questionnaire published by Craig et al. (2003). Instead, information on the validity and reliability of the Portuguese version shall be included.

It is also not mentioned on data curation. Did the authors follow the Guidelines of the IPAQ committee? Where there any data excluded from the further analysis concerning IPAQ methodology?

5.       The results section is clear. The presentation of the results is well prepared. In the discussion contributions of other studies that have looked at essentially the same topic of research are provided. The limitations of the study are also presented. In the conclusions the authors provide a general interpretation of the results in the context of other evidence and future research.

The reviewer suggests that the paper is valuable and worth to publish after minor corrections.

Reviewer 3 Report

First of all, I would like to thank you for the opportunity to review a paper with such an important and interesting topic as the one the authors present in their article.

The introduction is in line with the objective presented in the research.

The article presents an accurate approach in the statistical analysis used, although there is a limitation in the data collection format, as indicated in the limitations of the study by the authors. In this section, it would be interesting for the authors to indicate the Cronbach's Alpha value of the items of the sociodemographic questionnaire, or if another test was used to determine the level of reliability of the questionnaire, to indicate which one was used and the result obtained.

In the discussion section, it would be interesting to include studies reflecting changes in habits in relation to physical activity, sedentary activities or even eating habits during periods of pandemic or lockdown. There are different studies on this in different age groups.

I reiterate my thanks for the opportunity to review this work.
